# Learning Co-Speech Gesture for Multimodal Aphasia Type Detection

**Daeun Lee**[1*] **Sejung Son**[1*] **, Hyolim Jeon**[1,2], **Seungbae Kim**[3], **Jinyoung Han**[1,2†]

[1]Department of Applied Artificial Intelligence, Sungkyunkwan University, Seoul, South Korea
[2]Department of Human-AI Interaction, Sungkyunkwan University, Seoul, South Korea
[3]Department of Computer Science & Engineering, University of South Florida, Tampa, FL, USA
{delee12, jinyounghan}@skku.edu
{maze0717, gyfla1512}@g.skku.edu, seungbae@usf.edu

## Abstract

Aphasia, a language disorder resulting from brain damage, requires accurate identification of specific aphasia types, such as Broca's and Wernicke's aphasia, for effective treatment. However, little attention has been paid to developing methods to detect different types of aphasia. Recognizing the importance of analyzing co-speech gestures for distinguish aphasia types, we propose a multimodal graph neural network for aphasia type detection using speech and corresponding gesture patterns. By learning the correlation between the speech and gesture modalities for each aphasia type, our model can generate textual representations sensitive to gesture information, leading to accurate aphasia type detection. Extensive experiments demonstrate the superiority of our approach over existing methods, achieving state-of-the-art results (F1 84.2%). We also show that gesture features outperform acoustic features, highlighting the significance of gesture expression in detecting aphasia types. We provide the codes for reproducibility purposes[1].

## 1 Introduction

*Aphasia* is a language disorder caused by brain structure damage affecting speech functions, commonly triggered by stroke and other factors such as brain injury, dementia, and mental disorder (Broca et al., 1861; Wasay et al., 2014). Depending on prominent symptoms and severity, aphasia can be classified into the eight types (Kertesz, 2007) such as Broca's and Wernicke's aphasia as shown in Figure 5 in Appendix. People with aphasia (PWA) can face various communication challenges due to limited language processing and comprehension capabilities, which can result in difficulties with social interaction (El Hachioui et al., 2017). Since

---

*Equal contribution.
†Corresponding author.
[1]Code: https://github.com/DSAIL-SKKU/Multimod al-Aphasia-Type-Detection_EMNLP_2023

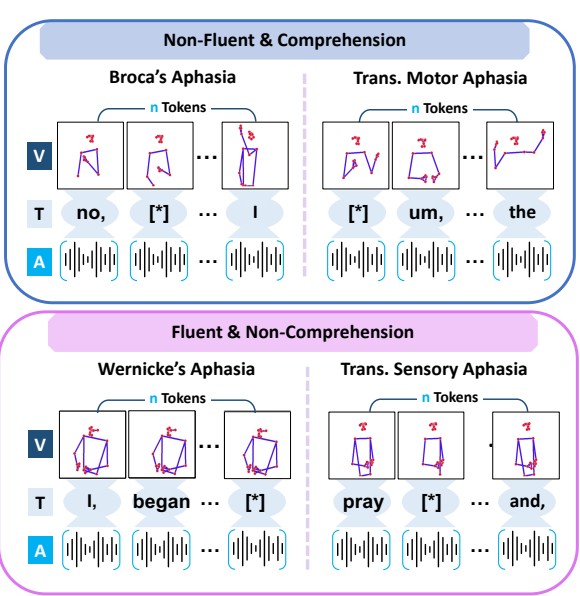

Figure 1: Variations in gestures observed across different types of aphasia. Each aligned data (text, audio, gesture) is extracted using Automatic Speech Recognition (ASR) (§3).

the traditional assessments for PWA are known to be time-consuming and costly (Wilson et al., 2018), identifying the pathological language impairment from speech data has received great attention (Qin et al., 2018a; Fraser et al., 2014). Clinically, a detailed diagnosis of aphasia type is imperative for proper treatment procedures; however, little attention has been paid to developing a classification model for the types of aphasia.

For diagnosing various types of aphasia, analyzing the co-speech gestures of PWA can be essential. PWA often rely on non-verbal communication techniques, especially gestures, as an additional communication tool due to difficulties in word retrieval and language errors (Preisig et al., 2018; de Kleine et al., 2023). Therefore, the same word can be interpreted differently depending on the accompanying gestures for different aphasia types. For example, as shown in Figure 1, individuals with Broca's aphasia show a higher frequency of iconic

gestures because Broca's area damage results in preserving semantic content but being impaired fluency (Albert et al., 1973). On the other hand, individuals with Wernicke's aphasia tend to derive limited assistance from gestures since fluent but incoherent speech pathology (Helm-Estabrooks, 2002). Hence, since the meaning of the word 'I' from PWA with Broca's aphasia may have more latent information, utilizing both speech (i.e., linguistic and acoustic) and gesture (i.e., visual) information is crucial in classifying aphasia types.

In combining two different modalities, speech and gesture, applying the existing multimodal fusion models is challenging due to the following two limitations. First, the existing models were usually built upon emotion recognition or sentiment analysis benchmarks (Zadeh et al., 2016; Busso et al., 2008) mainly consisting of facial information; yet, PWA struggles in using faces caused by the impaired emotion processing (Multani et al., 2017). Second, these models tended to capture a single link across the modalities, e.g., linking happiness-related words to smiling faces (Yang et al., 2021; Tsai et al., 2019), but there can be multiple links across the modalities depending on aphasia types. (e.g., 'I' may be connected to multiple gestures based on aphasia types, as shown in Figure 1).

To address these challenges, we propose to apply a graph neural network to generate multimodal features for each aphasia type, enabling the acquisition of rich semantics across modalities (Banarescu et al., 2013). Due to the heterogeneity among multimodalities in clinical datasets (Pei et al., 2023), a unique graph structure can help distinguish diverse patterns (Zhang et al., 2022) by learning the correlation between speech and the corresponding gestures of PWA with different aphasia.

Specifically, we construct a heterogeneous network that represents the relationships between speech (particularly including disfluency-related keywords and acoustic information) and gesture modalities based on their co-occurrence by using data from AphasiaBank (Forbes et al., 2012; MacWhinney et al., 2011), a shared database for aphasia research. Then, Speech-Gesture Graph Encoder (§4.2) extracts three node embeddings for disfluency-related keywords, audio, and gesture tokens by aggregating information from different sources. Before fusing the modalities, the Gesture-aware Word Embedding Layer (§4.3) generates textual representations sensitive to gesture informa-

tion by adjusting the weights of pre-trained word embeddings with the refined representations of disfluency tokens. Consequently, a multimodal Fusion Encoder (§4.4) is applied to incorporate aphasia-type-specific multimodal representations to predict the final aphasia type (§4.5).

The extensive experiments show that our model performs better than the prior methods in detecting aphasia types. We find that applying the GNN can effectively generate multimodal features by capturing relations between speech and gesture information, enhancing performance. Our analysis reveals that gesture features play more critical roles in predicting aphasia types than acoustic features, implying that gesture expression is a unique attribute of PWA with different types. We exemplify that the qualitative analysis based on the proposed model can help clinicians understand aphasia patients more comprehensively, helping to provide timely interventions. To the best of our knowledge, this is the first attempt that uses gesture and speech information for automatic aphasia types detection.

## 2 Related Work

**Aphasia analysis and detection.** To identify the pathological symptoms shown in a narrative speech of PWA, researchers have focused on linguistic features (e.g., word frequency, Part-of-Speech (Le et al., 2018), word embeddings (Qin et al., 2019a)), and acoustic features (e.g., filler words, pauses, the number of phones per word (Le and Provost, 2016; Qin et al., 2019b)). With the advances in automated speech recognition (ASR) that can make the transcription of aphasic speech into text (Radford et al., 2022; Baevski et al., 2020), there have been end-to-end approaches that do not require explicit feature extraction in assessing patients with aphasia (Chatzoudis et al., 2022; Torre et al., 2021). While these works reveal valuable insight into detecting aphasia, little attention has been paid to identifying the types of aphasia, which can be crucial for proper treatment procedures. Considering the importance of understanding gesture patterns for identifying the types of aphasia (Preisig et al., 2018), we propose a multimodal aphasia type prediction model using both speech and gesture information.

**Multimodal learning for detecting language impairment.** Multimodal language analysis helps to understand human communication by integrating information from multiple modalities for tasks such as sentiment analysis (Zadeh et al., 2017) and

emotion recognition (Yoon et al., 2022). However, previous studies simply concatenated features to identify language impairment in speech data (Rohanian et al., 2021; Balagopalan et al., 2020), which lacks comprehension of the complex connections between modalities (Cui et al., 2021; Syed et al., 2020). To capture interconnections between modalities, recent studies used Transformer-based multimodal models (Lin et al., 2022; Rahman et al., 2020a) that can extract contextual information across modalities, utilizing factorized co-attention (Cheng et al., 2021), self-attention (Hazarika et al., 2020), or cross-modal attention (Tsai et al., 2019). While the prior work tended to capture a single link between modalities, e.g., linking the 'happy' (linguistic modality) to a smiling face (visual modality) (Yang et al., 2021; Tsai et al., 2019), little attention has been paid to learning multiple links across the modalities, which can be crucial in aphasia type detection. As shown in Figure 1, speech and gesture modalities may have multiple links across the different aphasia types. Hence, to distinguish aphasia types, we propose to apply a graph neural network to generate multimodal features for each aphasia by learning the correlation between speech and gesture patterns of PWA.

## 3 Aphasia Dataset

**Data Collection** We sourced our dataset from the AphasiaBank (MacWhinney et al., 2011; Forbes et al., 2012), a shared database used by clinicians for aphasia research; the corpus information is summarized in Table 6 in Appendix. The dataset provides video recordings of the language evaluation test process between a pathologist and a subject, which also includes human-annotated transcriptions and subjects' demographic information. Among the evaluation tests, we chose the Cinderella Story Recall task (Bird and Franklin, 1996), where pictures from the Cinderella storybook are shown to participants, who are then asked to recall and retell the story spontaneously. The test has been proven as a helpful tool that can offer valuable insights into a subject's speech and language skills (Illes, 1989).

Each data is categorized into one of four types: *Control, Fluent, Non-Comprehension, Non-Fluent*, based on the ability to auditory comprehension and fluency skills, as shown in Table 1. Due to the limited amount of data available for training, specific types of aphasia, such as Global aphasia and Transcortical mixed aphasia, are excluded during

Table 1: Summary of data statistics of the aligned modalities with 50 tokens for each user. (✓: impaired, ✗: not impaired).

| Labels | Impaired | | Aphasia Types | Data Statistics | | |
| --- | --- | --- | --- | --- | --- | --- |
| | Flu. | Com. | | Subj. | Samp. | Dur. (s) |
| Ctrl | ✗ | ✗ | Control | 194 | 1,815 | 17.14 |
| Flu | ✗ | ✗ | Anomic | 143 | 927 | 31.80 |
| | | | Conduction | 62 | 424 | 26.27 |
| Non Com. | ✗ | ✓ | Sensory | 1 | 3 | 15.83 |
| | | | Wernicke | 26 | 188 | 24.68 |
| Non Flu. | ✓ | ✗ | Motor | 8 | 23 | 43.01 |
| | | | Broca | 73 | 271 | 34.75 |
| Total | | | | 507 | 3,651 | 23.78 |

data preprocessing.

**Data Preprocessing** We utilize Whisper (Radford et al., 2022), a popular ASR model, to generate automated transcriptions. Unlike previous ASR systems focusing on transcribing clean speech (Torre et al., 2021), Whisper can capture filler words (e.g., um, uh) and unintelligible disfluency words (e.g., [*]) at the token level. As depicted in Table 8 in Appendix, the Word Error Rate (WER) results show that the average WER for all participants is comparable to the previous research (Weiner et al., 2017). Furthermore, our research reveals that the text preprocessing results in a decrease in WER due to the frequent presence of fillers and disfluencies in the text of PWA, except for those with non-fluent aphasia types (e.g., *Non-Fluent*).

We then align the text tokens with corresponding gesture and audio tokens using the provided timestamps (Louradour, 2023). We extract the first frame (1 FPS) as a representative image for the gesture tokens. To augment the dataset, we chunk the aligned modalities with 50 tokens for each user except any segment less than 3 seconds long, as shown in Figure 1. Note that we exclude the segments of the interviewer in the video recordings since their subjective reactions can potentially affect the patient's performance (Parveen and Santhanam, 2021; Qin et al., 2018b).

Finally, the dataset consists of 507 subjects and their 3,651 aligned modalities, with an average duration of 23.78 seconds, as shown in Table 1.

## 4 Aphasia Type Detection

Figure 2 illustrates the overall architecture of the proposed model for detecting aphasia types. The model comprises four main components: Speech-Gesture Graph Encoder (§4.2), Gesture-aware Word Embedding Layer (§4.3), Multimodal Fusion Encoder (§4.4), and Aphasia Type Prediction Decoder (§4.5).

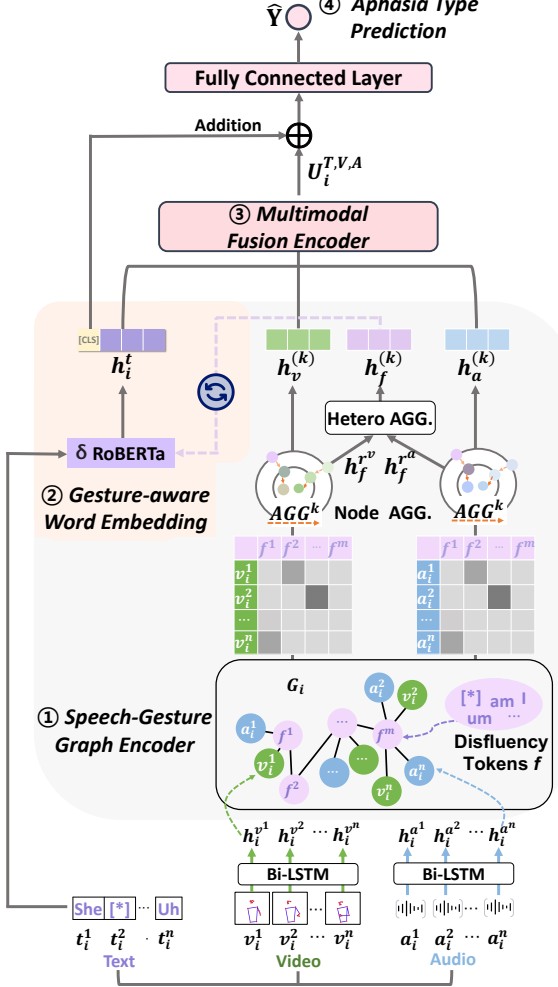

Figure 2: The overall architecture of the proposed model: ① Speech-Gesture Graph Encoder (§4.2), ② Gesture-aware Word Embedding Layer (§4.3), ③ Multimodal Fusion Encoder (§4.4), and ④ Aphasia Type Prediction Decoder (§4.5).

## 4.1 Problem Statement

Suppose we have a set of aphasia dataset $C = \{c_i\}_{i=1}^{|C|}$, $c_i$ is structured on an $n$ length of sequences, including text tokens ($t_i^n$), gesture tokens ($v_i^n$), and audio tokens ($a_i^n$) as described in Figure 1, which can be represented as $c_i = (t_i^n, v_i^n, a_i^n)$. Then, the proposed model is defined as a multi-class classification problem that classifies an aphasia data $c_i$ into an Aphasia type $y_i \in \{Control, Fluent, Non\text{-}Comprehension, Non\text{-}Fluent\}$.

## 4.2 Speech-Gesture Graph Encoder

We propose a Speech-Gesture Graph Encoder that can generate multimodal features specific to each aphasia type by learning the correlation between speech and gesture patterns in identifying the aphasia types.

### 4.2.1 Heterogeneous Graph Construction

In a preliminary analysis, we observed statistical differences between words, especially disfluency-related keywords, and aligned gesture features across different aphasia types (see Table 11 in Appendix). Based on the findings, we construct a set of heterogeneous graphs $\mathbb{G} = \{G_1, G_2, \ldots, G_i\}$ for $c_i$ that represents the relationships between disfluency-related keywords and multimodalities (i.e., gesture, audio) based on their co-occurrence. Note that we use the same disfluency tokens for each graph in $\mathbb{G}$ to capture the differences between users with different aphasia types. Disfluency keywords are extracted based on their frequency in ASR transcriptions, as shown in Table 9 in Appendix. Unlike previous approaches (Day et al., 2021), we do not remove 'stop words' because most disfluency-related words are associated with such 'stop words,' which are typically filtered out in text classification tasks. Formally, each graph is defined with the set of disfluency nodes $V_f$, which are connected to aligned gesture $V_v$ and audio $V_a$ nodes in aligned multimodal data $c_i$ as follows.

$$G_i = (V_f, V_v, V_a, E_{fv}, E_{fa}) \qquad (1)$$

**Textual Representation.** In order to extract the representation of disfluency node in $V_f$, we use the pre-trained RoBERTa (Liu et al., 2019) that is a pre-training language model shows a robust performance across various NLP tasks. After tokenizing the set of words $\{f_i^m\}_{i=1}^{|m|}$ into $e_i^f$, we encode them as follows.

$$h^f = RoBERTa(e^f) \in \mathbb{R}^{m \times d_t} \qquad (2)$$

where $d_t$ is the dimension of the textual feature. If any disfluency keywords are not included in the vocabulary of the pre-trained tokenizer, we add new tokens to represent these keywords.

**Visual Representation.** To utilize the gesture information for gesture nodes $V_v$, we extract the pose landmarks using MediaPipe[2] (Lugaresi et al., 2019) API that generates body pose landmarks, providing image-based 3-dimensional coordinates. Since our video data captures participants seated, we exclude the lower body keypoints and leverage the 23 upper body keypoints. Visual feature $e_i^v$ is fed into a bidirectional LSTM to derive context representation, reflecting the dynamic nature of an individual's gestures. Finally, the hidden state vectors are concatenated as follows.

---

[2]https://developers.google.com/mediapipe

$$h_i^v = \left[ \overrightarrow{LSTM} \left( e_i^v, h_{i-1}^v \right), \overleftarrow{LSTM} \left( e_i^v, h_{i+1}^v \right) \right] \quad (3)$$

**Acoustic Representation.** For audio feature extraction, we use OpenSmile (Eyben et al., 2010), an open-source audio processing toolkit, with extended Geneva Minimalistic Acoustic Parameter Set (eGeMAPS) (Eyben et al., 2015). We extract 25 low-level acoustic descriptors (LLDs) in each second, including loudness, MFCCs (Mel-frequency cepstral coefficients), and other relevant features. We then average the values of all 25 features to obtain audio features $e_i^a$. After that, the audio hidden state vectors are extracted, adopting the same approach for extracting visual features as follows.

$$h_i^a = \left[ \overrightarrow{LSTM} \left( e_i^a, h_{i-1}^a \right), \overleftarrow{LSTM} \left( e_i^a, h_{i+1}^a \right) \right] \quad (4)$$

### 4.2.2 Cross-relation Aggregation

To update the features of the target node $V$, we adopt GraphSAGE (Hamilton et al., 2017), a widely used GNN model that supports batch training without requiring updates across the entire graph. This recursive approach involves updating the embeddings of each node by aggregating information from its immediate neighbors using an aggregation function at each search depth ($k$). Specifically, the representation $h_V^k$ of node $V$ is updated by combining $h_V^{k-1}$ with information obtained from $h_{\mathcal{N}(V)}^{(k)}$, which represents the neighboring nodes of $V$ at step $k$. The initial output is $h_V^0 = h_V$, and the series of updating processes are defined as follows.

$$h_{\mathcal{N}(V)}^{(k)} = \text{NodeAGG}^k \left( \{ h_u^{k-1}, \forall u \in \mathcal{N}(V) \} \right) \quad (5)$$

$$h_V^{(k)} = \sigma \left( W^k \cdot (h_V^{k-1} \oplus h_{\mathcal{N}(V)}^k) \right) \quad (6)$$

If the target node $V$ has multiple relations from different types of source nodes $\{r_j\}_{j=1}^{|N(v)|} \in N(v)$, $V$ has heterogeneous representations as follows.

$$h_V^{(k)} = [h_V{}^{r_1}, h_V{}^{r_2}, \cdots, h_V{}^{r_j}] \quad (7)$$

where $j$ is the number of source node types among the neighbors of target node $V$. Since latent information from distinct source node embeddings can affect the target node representation differently, we aim to aggregate the fine-grained interplay between them without losing valuable knowledge by maintaining independent distributions. Finally, we derive a feature of node $V$ at step $k$ as follows.

$$\acute{h}_V^{(k)} = \text{HeteroAGG}^K \left( h_V^{(k)} \right) \quad (8)$$

In this way, three aggregated representations for disfluency, gesture, and audio tokens, respectively, are obtained as follows.

$$H_i = \{ \acute{h}_f^{(k)}, \acute{h}_a^{(k)}, \acute{h}_v^{(k)} \} \quad (9)$$

### 4.3 Gesture-aware Word Embedding Layer

To obtain textual features $h_i^t$ sensitive to gesture information, we update the pre-trained RoBERTa word embedding weights (Liu et al., 2019) with updated disfluency representations $\acute{h}_f^{(k)}$ as follows.

$$\acute{h}_f^{(k)} = \delta RoBERTa(e^f) \in \mathbb{R}^{mXd_t} \quad (10)$$

$$h_i^t = \delta RoBERTa(e_i^t) \in \mathbb{R}^{nXd_t} \quad (11)$$

Subsequently, the final multimodal representation for $c_i$ is derived as follows.

$$\acute{H}_i = \{ h_i^t, \acute{h}_a^{(k)}, \acute{h}_v^{(k)} \} \quad (12)$$

### 4.4 Multimodal Fusion Encoder

To fuse multimodal representations, we utilize a Multimodal Transformer (Tsai et al., 2019) that effectively integrates multimodal features by capturing contextual information. We employ two cross-attention layers and a self-attention mechanism for generating each modality feature as shown Figure 7 in Appendix. For example, when the model generates a textual feature, each layer (i.e., (V→T), (A→T)) uses the visual and acoustic representations as the key/value pairs, respectively, and textual representation as the query vector. After concatenating two features from each layer, a self-attention Transformer fuses them to generate cross-modal representation $U^T$. Next, the model concatenates three cross-modal features (i.e., $U^T, U^V, U^A$) to derive the final multimodal representation $U^{TVA}$ as follows (See Tsai et al. (2019) for more details).

$$U_i^{TVA} = U_i^T \oplus U_i^V \oplus U_i^A \quad (13)$$

Finally, we add the pooled embeddings of [CLS] token in $h_i^t$ and the $U^{TVA}$ as follows.

$$\acute{U}_i = U_i^{TVA} + h_i^{t^{CLS}} \quad (14)$$

### 4.5 Aphasia Type Prediction

To predict the types of aphasia for $c_i$, the final prediction vector is generated as follows.

$$\hat{y} = \mathcal{F}(ReLU(\mathcal{F}(\acute{U}_i))) \quad (15)$$

where $\mathcal{F}$ is a fully-connected layer and $ReLU$ is an activation function. Finally, the cross-entropy

loss is calculated using the probability distribution $y$ and classification score $\hat{y}$ obtained as follows.

$$\mathcal{L} = -\frac{1}{b}\sum_{i=1}^{b} y_i \log \hat{y}_i \qquad (16)$$

where $b$ is the batch size.

# 5 Experiments

## 5.1 Experimental Settings

Table 10 in Appendix shows that we split the dataset into the train and test sets with an 8:2 ratio. In all our experiments, we ensure that users in the test set are entirely disjoint and do not overlap with those in the training set. For reproducibility, detailed experimental settings are summarized in Appendix B.1.

## 5.2 Baselines

To conduct extensive performance comparisons, we consider two categories of baseline methods: (i) aphasia & dementia detection models and (ii) multimodal fusion baselines. A detailed explanation of the baselines is summarized in Appendix B.2.

**Aphasia & Dementia Detection Baselines.** Since predicting aphasia types has yet to be explored, we compare the approaches from the related tasks that only detect the presence or absence of aphasia. Additionally, we compare the popular models for dementia detection: (i) Logistic Regression (LR) (Cui et al., 2021; Syed et al., 2020), (ii) Support Vector Machines (SVM) (Qin et al., 2018b; Rohanian et al., 2021), (iii) Random Forest (RF) (Cui et al., 2021; Balagopalan et al., 2020), (iv) Decision Tree (DT) (Qin et al., 2018b; Balagopalan and Novikova, 2021), and (v) AdaBoost (Cui et al., 2021).

**Multimodal Fusion Baselines.** We compare the proposed model with the following four existing multimodal fusion methods: (i) MulT (Tsai et al., 2019), (ii) MISA (Hazarika et al., 2020), (iii) MAG (Rahman et al., 2020b), and (iv) SP-Transformer (Cheng et al., 2021).

# 6 Results

## 6.1 Model Performance

Table 2 shows the weighted average F1-score of the baselines and the proposed model across the aphasia types. The proposed model outperforms all the baselines (F1 84.2%) regardless of the aphasia types. Specifically, deep learning-based multimodal fusion models perform better than models that simply concatenate multimodal features in

Table 2: Comparison of performance between the proposed model and baseline models, with results averaged over a Group Stratified 5-fold cross-validation. Results marked with an asterisk (*) indicate statistical significance compared to MAG ($p < 0.05$) according to the Wilcoxon's signed rank test.

| Model | | Label (F1-score) | | | | |
| --- | --- | --- | --- | --- | --- | --- |
| | | Total | Ctrl | Flu | Non Com. | Non Flu. |
| Aphasia& Dementia Detection | SVM | 0.338 | 0.670 | 0.000 | 0.000 | 0.000 |
| | RF | 0.742 | 0.871 | 0.719 | 0.000 | 0.400 |
| | DT | 0.688 | 0.824 | 0.652 | 0.090 | 0.283 |
| | LR | 0.655 | 0.773 | 0.696 | 0.000 | 0.000 |
| | AdaBoost | 0.719 | 0.882 | 0.662 | 0.000 | 0.303 |
| Fusion | MISA | *0.761* | 0.899 | 0.741 | 0.000 | 0.349 |
| | MulT | 0.761 | 0.885 | *0.750* | 0.000 | 0.410 |
| | MAG | 0.725 | 0.838 | 0.698 | 0.000 | *0.514* |
| | SP-Trans. | 0.756 | 0.893 | 0.742 | 0.000 | 0.324 |
| **Ours** | 30 Tok. | 0.732 | *0.906* | 0.676 | **0.303** | 0.446 |
| | **50 Tok.** | **0.842\*** | **0.949\*** | **0.840\*** | *0.125* | **0.530** |

Table 3: Ablation study to examine the effectiveness of §4.2 Speech-Gesture Graph Encoder and §4.4 Multimodal Fusion Encoder.

| Component | | Prec | Rec | F1 |
| --- | --- | --- | --- | --- |
| Node agg. | LSTM | 0.777 | 0.790 | 0.780 |
| | Mean | 0.722 | 0.743 | 0.722 |
| | Pool | 0.781 | 0.807 | 0.787 |
| | **BiLSTM** | **0.837** | **0.852** | **0.842** |
| Hetero agg. | Mean | 0.785 | 0.763 | 0.771 |
| | Sum | 0.759 | 0.774 | 0.761 |
| | Max | 0.794 | 0.803 | 0.783 |
| | **Min** | **0.837** | **0.852** | **0.842** |
| Fusion | Concat | 0.746 | 0.609 | 0.568 |
| | Multiply | 0.776 | 0.808 | 0.784 |
| | Add | 0.789 | 0.793 | 0.751 |
| | SP-Trans. | 0.757 | 0.781 | 0.758 |
| | **MulT** | **0.837** | **0.852** | **0.842** |

predicting aphasia types. We observe that the proposed model can relatively accurately predict minorities such as $Non-Comprehension$ (N=191) than other baselines. We also investigate how the number of tokens $n$ decided in creating an aligned multimodal dataset (Figure 1) affects the model performance. As demonstrated in Table 2, we find that the F1-score is higher for the model using *50 tokens* than *30 tokens*, which indicates the importance of comprehending longer sequences for accurate prediction of aphasia types. While the utilization of *30 tokens* improves the performance of $Non-Comprehension$, we attribute this improvement to the larger size of the dataset based on *30 tokens*, as indicated in Table 7 in the Appendix.

## 6.2 Ablation Study

We perform an ablation study to examine the effectiveness of each component of the proposed model.
**Analysis on Model Components.** As described

in Table 3, the model performance significantly improves when we use the BiLSTM node aggregator (Tang et al., 2020). Given that our dataset is structured on a sequence level, RNN-based models exhibit higher performance with their capacity to capture long-term dependencies. Also, the Min heterogeneous aggregate function (Wang et al., 2019) can effectively learn relationships between sources from multiple modalities. We also compare the multimodal fusion models, including Transformer-based multimodal fusion models (Tsai et al., 2019; Cheng et al., 2021) as a Multimodal Fusion Encoder. As shown in Table 3, MulT (Tsai et al., 2019) model performs better at 84.2% of F1-score, suggesting its capability to generate a more informative multimodal representation.

**Analysis on Different Modalities.** To analyze the significance of each modality in detecting aphasia types, we perform an analysis of unimodal models trained with each modality. For unimodal aphasia type detection, we solely utilized the unimodal Transformer encoder. As shown in Table 4, the model trained with text features performs better (F1 70.0%). This indicates that linguistic features are more informative for identifying language impairment disorders (Cui et al., 2021; Chen et al., 2021). Additionally, visual features outperform acoustic features (F1 62.9%), suggesting that PWA exhibit distinct characteristics in their gesture expression. When constructing a graph $G$, employing only the edge $E_{fv}$ between disfluency and gesture tokens results in higher performance (F1 77.3%) compared to using the edge $E_{fa}$ between disfluency and audio tokens (F1 74.7%), as described in Table 4. However, considering all three modalities together shows the best performance, which reveals that learning linguistic features along with visual and acoustic characteristics and their relations is more effective than relying solely on one modality.

**Analysis on Speech-Gesture Graph Encoder.** We next explore the effectiveness of the proposed Speech-Gesture Graph Encoder to understand how the multimodal features specific to each aphasia type are helpful in performance. We first find that using multimodal features from the graph encoder helps to improve prediction performance (F1 79.2%) compared to without graph encoder (F1 76.5%), as described in Table 4. We attribute this to the strength of the graph neural network model, which can learn better representations from cross-modal relations. Also, the performance notably

Table 4: Comparison of the impact of §4.2 Speech-Gesture Graph Encoder and §4.3 Gesture-aware Word Embedding Layer on each modality.

| Component | | | Metrics | | |
|---|---|---|---|---|---|
| Graph Encoder | Update Embed. | Modality | Prec | Rec | F1 |
| ✗ | ✗ | Text | 0.692 | 0.727 | 0.700 |
| ✗ | ✗ | Acoustic | 0.455 | 0.566 | 0.501 |
| ✗ | ✗ | Visual | 0.591 | 0.689 | 0.629 |
| ✗ | ✗ | T+V+A | 0.749 | 0.786 | 0.765 |
| ✓ | ✗ | T+V+A | 0.774 | 0.814 | 0.792 |
| ✓ | ✓ | T+V | 0.767 | 0.781 | 0.773 |
| ✓ | ✓ | T+A | 0.771 | 0.740 | 0.747 |
| ✓ | ✓ | **T+V+A** | **0.837** | **0.852** | **0.842** |

improves when using refined word embeddings (F1 84.2%) compared to just pre-trained word embeddings (F1 79.2%). This implies that gesture-sensitive text features can help the multimodal fusion encoder to assign different weights to each modality considering the meaning of words.

We further conduct experiments to determine the optimal number $m$ of disfluency token nodes $V_f$ in the graph $G$, ranging from 50 to 300. Figure 3 illustrates the weighted average F1/Precision/Recall scores for identifying aphasia types across the different $m$ disfluency keywords. The performance improves as more keywords are included, but no further enhancement is observed beyond 150 keywords. However, our analysis demonstrates that using 100 keywords produces better results compared to using 300 keywords. We believe that extracting important disfluency keywords will result in improved outcomes. In our future work, we plan to collaborate with clinical experts, including speech-language pathologists and neurologists, to acquire valuable clinical insights and guidance.

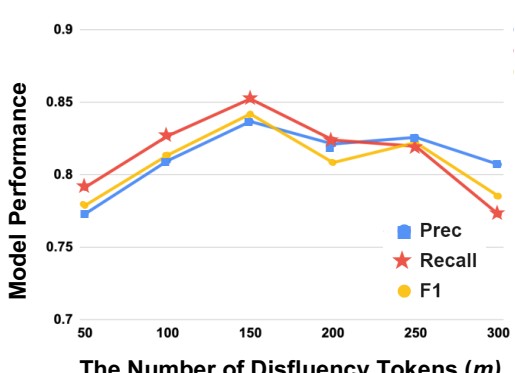

Figure 3: Performance of the model by the number of disfluency tokens ($m$).

**Analysis on Different Gender.** Earlier clinical research indicates that different gender is associated with aphasia severity, including communica-

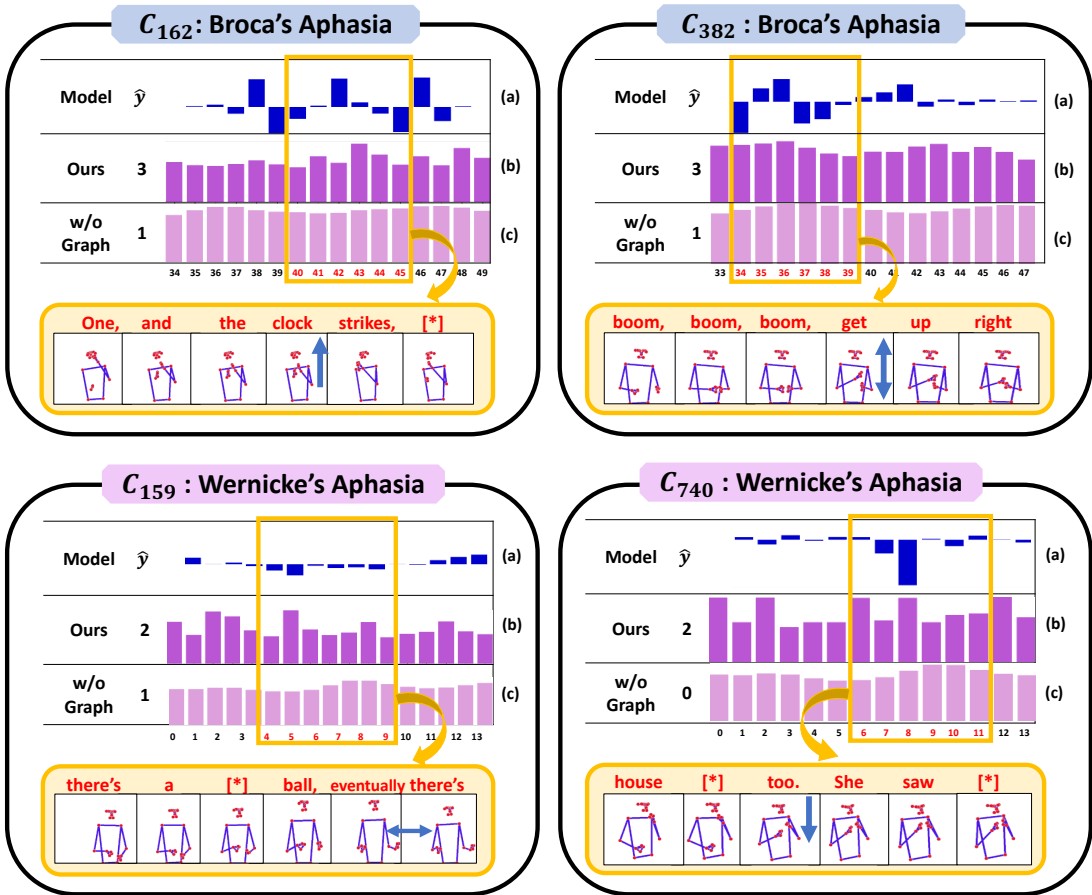

Figure 4: Visualization of the crossmodal attention matrix from the V → L network of the Multimodal Fusion Encoder for the proposed model with (b) or without (c) the speech-gesture graph encoder. (a) presents the differences in the spatial position of the right wrist's landmark compared to the previous frame, where positive and negative values indicate upward/rightward and downward/leftward movements.

Table 5: Gender differences in the model performance.

| Train | Test | Prec | Rec | F1 |
|---|---|---|---|---|
| Both | Both | **0.837** | **0.852** | **0.842** |
| | Female | 0.889 | 0.890 | 0.885 |
| | Male | 0.795 | 0.820 | 0.805 |
| Female | Both | 0.718 | 0.761 | 0.724 |
| | Female | 0.757 | 0.801 | 0.772 |
| | Male | 0.668 | 0.727 | 0.683 |
| Male | Both | 0.795 | 0.771 | 0.777 |
| | Female | 0.805 | 0.758 | 0.773 |
| | Male | 0.794 | 0.782 | 0.785 |

tion impairment and lower scores on specific subtests (Sharma et al., 2019). To validate this observation, we train and validate our model using separate datasets of either male or female participants (Table 5). The model performs better when trained on the male dataset (F1 77.7%) than the female dataset (F1 72.4%), despite a larger training size from females (Table 10). These findings suggest that the linguistic impairment caused by aphasia is more pronounced in male patients, as reported in previous studies (Yao et al., 2015). Thus, we believe the model can learn linguistic impairment markers better when trained with the male dataset.

## 6.3 Qualitative Analysis

In this section, we perform a qualitative analysis on four representative cases involving different types of aphasia, especially Broca's (i.e., $c_{162}$, $c_{382}$) and Wernicke's aphasia (i.e., $c_{159}$, $c_{740}$). As shown in Figure 4, we compare the cross-modal attention matrix from the V → L network of the Multimodal Fusion Encoder in the proposed model with or without the speech-gesture graph encoder. Each attention weight can be interpreted as the relevance of the visual and the textual features. Note that Figure 6 in Appendix visualizes the standard deviation of pose landmarks of individuals, which reveals that the Non-Fluent type (e.g., Broca's aphasia) exhibits a broader range of activity compared to the Non-Comprehension type (e.g., Wernicke's aphasia) (Lanyon and Rose, 2009).

We find that the model without a graph encoder assigns the same attention regardless of the aphasia types when significant physical actions are displayed. However, our proposed model can predict Broca's aphasia (i.e., $c_{162}$, $c_{382}$) by assigning higher attention weights if meaningful motions are cap-

tured. For example, $c_{162}$ points the index finger to express the *'clock'* and $c_{382}$ indicates upward movements during phrases like *'get up'*. This implies that PWA with Broca's aphasia leverage gestures as an additional means of communication (Preisig et al., 2018; de Kleine et al., 2023), and the proposed model can accurately recognize the meaning of gestures associated with the text. By contrast, in the cases of Wernicke's aphasia ($c_{159}, c_{740}$), where motions are commonly small, the model assigns higher attention scores to any noticeable motion change, despite the meaning of the movement.

We believe the proposed model with a speech-gesture graph encoder can effectively generate aphasia type-specific multimodal features for predicting aphasia types, as demonstrated in our case study. Thus, the proposed model can be used for screening and identifying individuals with aphasia to prioritize early intervention for clinical support.

## 7 Conclusion

In this paper, our approach utilizing GNNs proves effective in generating multimodal features specific to each aphasia type for predicting aphasia types. The extensive experiments demonstrated state-of-the-art results, with visual features outperforming acoustic features in predicting aphasia types. We plan to provide the codes for reproducibility, facilitating further research in this area. The qualitative analysis offered valuable insights that can enhance clinicians' understanding of aphasia patients, leading to more comprehensive assessments and timely interventions. Overall, our research contributes to the field of aphasia diagnosis, highlighting the importance of using multimodal information and graph neural networks. We believe our findings will drive advancements in speech and language therapy practices for individuals with aphasia.

## Ethics Statement

We sourced the dataset from AphasiaBank with Institutional Review Board (IRB) approval [3] and strictly follow the data sharing guidelines provided by TalkBank [4], including the Ground Rules for all TalkBank databases based on American Psychological Association Code of Ethics (Association et al., 2002). Additionally, our research adheres to the five core issues outlined by TalkBank in accordance with the General Data Protection Regulation

---

[3]SKKU2022-11-039

[4]https://talkbank.org/share/ethics.html

(GDPR) (Regulation, 2018). These issues include addressing commercial purposes, handling scientific data, obtaining informed consent, ensuring deidentification, and maintaining a code of conduct. Considering ethical concerns, we did not use any photographs and personal information of the participants from the AphasiaBank.

## Limitations

First, traditional aphasia classification schemes, such as the WAB (Western Aphasia Battery) (Kertesz, 2007), have faced criticism because patients' symptoms often do not fit into a single type, and there is overlap between the different classes (Caramazza, 1984; Swindell et al., 1984). However, the WAB still offers a means to categorize patients based on their most prominent symptoms. Furthermore, datasets labeled using the WAB scheme are essential for studies that rely on methods requiring a substantial amount of training data.

Second, the previous study highlighted the potential of identifying repetitive mouth patterns to detect speech impairments (Einfalt et al., 2019). However, capturing mouth landmarks on the subjects was challenging due to the considerable distance at which the videos were recorded in the AphasiaBank dataset. Considering the significant role of mouth information in human communication (Busso et al., 2004), we anticipate that incorporating mouth information in conjunction with gesture information will improve performance.

Additionally, we utilized computational methods to extract disfluency-related keywords from frequent occurrences, which were then used for constructing the graph. However, our analysis in Figure 3 demonstrated that using 100 tokens produced better results compared to using 300 tokens. Consequently, we believe that extracting significant disfluency tokens guided by expert advice will result in improved outcomes. In our future work, we intend to collaborate with clinical experts, including speech-language pathologists and neurologists, to acquire valuable clinical insights and guidance.

## Acknowledgments

This research was supported by the Ministry of Education of the Republic of Korea and the National Research Foundation (NRF) of Korea (NRF-2022S1A5A8054322) and the National Research Foundation of Korea grant funded by the Korea government (MSIT) (No. 2023R1A2C2007625).

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

## A Dataset

### A.1 AphasiaBank Dataset

As illustrated in Table 6, we sourced our dataset from the AphasiaBank (MacWhinney et al., 2011; Forbes et al., 2012), a shared multimedia database used by clinicians for aphasia research [5].

Table 6: The corpora we used from AphasiaBank

| Corpus | Site |
|---|---|
| ACWT (Bynek, 2013) | Aphasia Center of West Texas |
| Adler (Szabo, 2013) | Adler Aphasia Center |
| APROCSA (Wilson, 2021) | Vanderbilt University Medical Center |
| BU (Hoover, 2013) | Boston University |
| Capilouto (Capilouto, 2008) | University of Kentucky |
| CC-Stark (Stark, 2022) | file for CC |
| CMU (MacWhinney, 2013) | Carnegie Mellon University |
| Elman (Elman, 2011, 2016) | Aphasia Center of California |
| Fridriksson (Fridriksson, 2013) | University of South Carolina |
| Garrett (Garrett, 2013) | Pittsburgh, PA |
| Kansas (Jackson, 2013) | University of Kansas |
| Kempler (Kempler, 2013) | Emerson College |
| Kurland (Kurland, 2013) | University of Massachusetts, Amherst |
| MSU (Boyle, 2013) | Montclair State University |
| NEURAL (Stark, 2023) | NEURAL Research Lab, Indiana University |
| Richardson (Richardson, 2008) | University of New Mexico |
| SCALE (McCall, 2013) | Snyder Center for Aphasia Life Enhancement |
| STAR (Corwin, 2013) | Stroke Aphasia Recovery Program |
| TAP (Silverman, 2013) | Triangle Aphasia Project |
| TCU (Muñoz, 2013) | Texas Christian University |
| Thompson (Thompson, 2013) | Northwestern University |
| Tucson (Hirsch Kruse, 2013) | University of Arizona |
| UCL (Dean, 2021) | University College London |
| UMD (Faroqi-Shah and Milman, 2018) | University of Maryland |
| UNH (Ramage, 2013) | University of New Hampshire |
| Whiteside (Whiteside, 2013) | University of Central Florida |
| Williamson (Williamson, 2013) | Stroke Comeback Center |
| Wozniak (Wozniak, 2013) | InteRACT |
| Wright (Wright, 2013) | Arizona State University |

Table 7: Comparison of the number of samples in 50 Token data and 30 Token data.

| Labels | Impaired | | Aphasia Types | # Sample | |
| | Flu. | Com. | | 50 tokens (ours) | 30 tokens |
|---|---|---|---|---|---|
| Ctrl. | ✗ | ✗ | Control | 1,815 | 3,157 |
| Flu. | ✗ | ✗ | Anomic | 927 | 1,639 |
| | | | Conduction | 424 | 753 |
| Non-Com. | ✗ | ✓ | Trans. Sensory | 3 | 5 |
| | | | Wernicke | 188 | 331 |
| Non-Flu. | ✓ | ✗ | Trans. Motor | 23 | 44 |
| | | | Broca | 271 | 518 |
| Total | | | | 3,651 | 6,447 |

Table 8: The Word Error Rate (WER) across Aphasia Types for all participants.

| Labels | Impaired | | Aphasia Types | Average WER | |
| | Flu. | Com. | | Raw | Pre-proc. |
|---|---|---|---|---|---|
| Ctrl. | ✗ | ✗ | Control | 0.332 | 0.18 |
| Flu. | ✗ | ✗ | Anomic Conduction | 0.714 | 0.623 |
| Non-Com. | ✗ | ✓ | Trans. Sensory Wernicke | 0.607 | 0.542 |
| Non-Flu. | ✓ | ✗ | Trans. Motor Broca | 0.967 | 0.974 |
| Total | | | | 0.615 | 0.521 |

[5] https://talkbank.org/share/citation.html

### A.2 Aphasia Types

As shown in Figure 5 in Appendix, aphasia can be classified into eight types based on the Western Aphasia Battery (WAB) (Kertesz, 2007), a standard protocol for categorizing patients based on the most prominent symptoms and severity, and they are next broken down by the ability to auditory comprehension and fluency skills.

### A.3 Disfluency Keywords

Table 9 shows the examples of disfluency keywords $f$.

Table 9: The examples of disfluency keywords $f$

| Disfluency Tokens |
|---|
| '[*]', 'the', 'and', 'to', 'um', 'she', 'a', 'her', 'was', 'they', 'so', 'that', 'of', 'cinderella', 'it', 'uh', 'in', 'i', 'he', 'but', 'all', 'is', 'had', 'then', 'ball', 'go', 'with', 'prince', 'this', 'one', 'on', 'you', 'two', 'there', 'were', 'for', 'going', 'king', 'be', 'know', 'out', 'at', 'no', 'not', 'like', 'slipper', 'get', 'have', 'dress', 'got', 'up', 'went', 'very', 'who', 'glass', 'because', 'or', 'time', 'oh', 'stepmother', 'into', 'do', 'shes', 'what', 'beautiful', 'back', 'its', 'home', 'girl', 'woman', 'fairy', 'when', 'godmother', 'would', 'said', 'just', 'as', 'has', 'little', 'shoe', 'well', 'dont', 'his', 'mother', 'them', 'daughters', 'house', 'midnight', 'stepsisters', 'by', 'men', 'bye', 'girls', 'sisters', 'mice', 'everything', 'fit', 'came', 'other', 'goes' |

### A.4 Analysis on Gesture Difference

We applied ANOVA to assess the differences in gesture features among the different aphasia types groups as well as the control group, as shown in Table 11. This analysis allowed us to determine whether there are significant differences in gesture patterns between the aphasia types.

## B Experiment

### B.1 Experimental Settings

As shown in Table 10, we split the dataset into the train and test sets with a 8:2 ratio. We tune hyperparameters based on the highest F1 score obtained from the cross-validation set for the models. We use the grid search to explore the dimension of hidden state $U_i \in \{64, 128, 256, 512, 768\}$, number of LSTM layers $n \in \{1, 2, 5\}$, dropout $\sigma \in \{0.1, 0.2, 0.3, 0.4, 0.5\}$, and initial learning rate $lr \in \{1e-5, 2e-5, 3e-5, 5e-5\}$. The optimal hyperparameters were found to be: $U_i = 768$,

Table 10: Gender distribution in the dataset.

| Gender | Train | Test | Total |
|---|---|---|---|
| All | 2,947 | 704 | 3,651 |
| Female | 1,633 | 327 | 1,960 |
| Male | 1,314 | 377 | 1,691 |

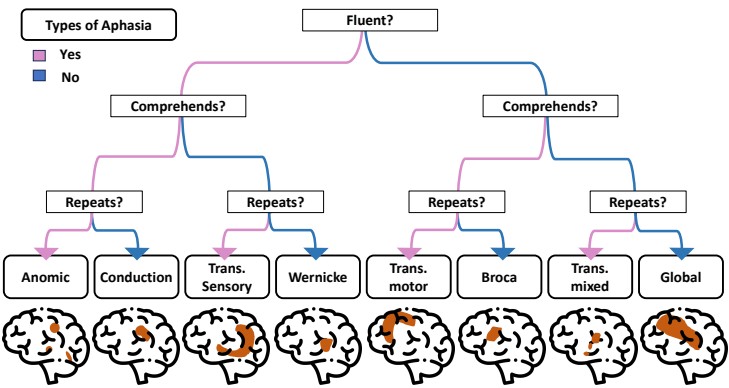

Figure 5: Types of Aphasia

Table 11: Comparison of gesture characteristics among different groups based on Aphasia types using ANOVA. The table below presents the p-values from comparing the gesture features of the top 10 tokens with the highest occurrence. (** means p-value < 0.001 / * means p-value < 0.05)

| Keypoints | [*] | the | and | to | um | she | a | her | was | they |
|---|---|---|---|---|---|---|---|---|---|---|
| NOSE | 0.000 ** | 0.000 ** | 0.000 ** | 0.0099 * | 0.000 ** | 0.000 ** | 0.000 ** | 0.000 ** | 0.000 ** | 0.2223 |
| LEFT_EYE_INNER | 0.000 ** | 0.000 ** | 0.000 ** | 0.0202 * | 0.000 ** | 0.000 ** | 0.000 ** | 0.000 ** | 0.000 ** | 0.3069 |
| LEFT_EYE | 0.000 ** | 0.000 ** | 0.000 ** | 0.000 ** | 0.016 * | 0.000 ** | 0.000 ** | 0.000 ** | 0.000 ** | 0.2201 |
| LEFT_EYE_OUTER | 0.000 ** | 0.000 ** | 0.000 ** | 0.000 ** | 0.0397 * | 0.000 ** | 0.000 ** | 0.000 ** | 0.000 ** | 0.1535 |
| RIGHT_EYE_INNER | 0.2061 | 0.000 ** | 0.0858 | 0.3253 | 0.000 ** | 0.1162 | 0.0143 * | 0.000 ** | 0.000 ** | 0.4163 |
| RIGHT_EYE | 0.000 ** | 0.000 ** | 0.0137 * | 0.2313 | 0.000 ** | 0.1287 | 0.0094 * | 0.000 ** | 0.000 ** | 0.243 |
| LEFT_EYE_OUTER | 0.000 ** | 0.000 ** | 0.000 ** | 0.0498 * | 0.000 ** | 0.0518 | 0.000 ** | 0.000 ** | 0.000 ** | 0.0819 |
| LEFT_EAR | 0.000 ** | 0.000 ** | 0.0614 | 0.7237 | 0.0562 | 0.0481 * | 0.6547 | 0.000 ** | 0.000 ** | 0.9854 |
| RIGHT_EAR | 0.000 ** | 0.000 ** | 0.000 ** | 0.000 ** | 0.000 ** | 0.000 ** | 0.000 ** | 0.000 ** | 0.000 ** | 0.000 ** |
| MOUTH_LEFT | 0.000 ** | 0.000 ** | 0.000 ** | 0.000 ** | 0.0057 * | 0.000 ** | 0.000 ** | 0.000 ** | 0.000 ** | 0.1522 |
| MOUTH_RIGHT | 0.1881 | 0.000 ** | 0.0958 | 0.2863 | 0.000 ** | 0.163 | 0.0185 * | 0.000 ** | 0.000 ** | 0.4538 |
| LEFT_SHOULDER | 0.000 ** | 0.000 ** | 0.000 ** | 0.000 ** | 0.3065 | 0.000 ** | 0.000 ** | 0.000 ** | 0.000 ** | 0.2222 |
| RIGHT_SHOULDER | 0.000 ** | 0.000 ** | 0.000 ** | 0.000 ** | 0.000 ** | 0.000 ** | 0.000 ** | 0.000 ** | 0.000 ** | 0.000 ** |
| LEFT_ELBOW | 0.000 ** | 0.000 ** | 0.000 ** | 0.000 ** | 0.6054 | 0.000 ** | 0.000 ** | 0.000 ** | 0.000 ** | 0.000 ** |
| RIGHT_ELBOW | 0.000 ** | 0.000 ** | 0.000 ** | 0.000 ** | 0.000 ** | 0.000 ** | 0.000 ** | 0.000 ** | 0.000 ** | 0.000 ** |
| LEFT_WRIST | 0.000 ** | 0.000 ** | 0.000 ** | 0.000 ** | 0.2786 | 0.000 ** | 0.000 ** | 0.000 ** | 0.000 ** | 0.000 ** |
| RIGHT_WRIST | 0.000 ** | 0.000 ** | 0.000 ** | 0.000 ** | 0.000 ** | 0.000 ** | 0.000 ** | 0.000 ** | 0.000 ** | 0.000 ** |
| LEFT_PINKY | 0.000 ** | 0.000 ** | 0.000 ** | 0.000 ** | 0.7116 | 0.000 ** | 0.000 ** | 0.000 ** | 0.000 ** | 0.000 ** |
| RIGHT_PINKY | 0.000 ** | 0.000 ** | 0.000 ** | 0.000 ** | 0.000 ** | 0.000 ** | 0.000 ** | 0.000 ** | 0.000 ** | 0.000 ** |
| LEFT_INDEX | 0.000 ** | 0.000 ** | 0.000 ** | 0.000 ** | 0.6041 | 0.000 ** | 0.000 ** | 0.000 ** | 0.000 ** | 0.000 ** |
| RIGHT_INDEX | 0.000 ** | 0.000 ** | 0.000 ** | 0.000 ** | 0.000 ** | 0.000 ** | 0.000 ** | 0.000 ** | 0.000 ** | 0.000 ** |
| LEFT_THUMB | 0.000 ** | 0.000 ** | 0.000 ** | 0.000 ** | 0.4492 | 0.000 ** | 0.000 ** | 0.000 ** | 0.000 ** | 0.000 ** |
| RIGHT_THUMB | 0.000 ** | 0.000 ** | 0.000 ** | 0.000 ** | 0.000 ** | 0.000 ** | 0.000 ** | 0.000 ** | 0.000 ** | 0.000 ** |

$n = 2$, $\sigma = 0.1$, and $lr = 1e - 5$. We implement all the methods using PyTorch 1.12 and optimize with the mini-batch AdamW (Loshchilov and Hutter, 2017) with a batch size of 32. We use the Exponential Learning rate Scheduler with gamma 0.001. We train the model on a GeForce RTX 3090 GPU for 50 epochs and apply early stopping with patience of 7 epochs.

### B.2 Baselines

**Aphasia & Dementia Detection Baselines.** Aphasia and dementia are neurological disorders that affect language and cognition, leading to difficulties in various cognitive functions and communication. Aphasia is caused by specific brain damage, resulting in language impairments, while dementia involves brain neuron degeneration, leading to overall cognitive decline and possible language impairments. Although both disorders impact language and cognition, a clear understanding of their differences and similarities is required for individual diagnosis and assessment.

- **LR** (Cui et al., 2021; Syed et al., 2020): We employed a Logistic Regression Classifier with an added L2 penalty term, using a cost parameter $c$ value of 0.01.

- **SVM+Poly** (Qin et al., 2018b; Rohanian et al., 2021): Each subject's text and audio embedding vector is fed to a Support Vector Machine with a polynomial kernel using the cost parameter $c = 0.01$.

Table 12: Performance comparisons of the proposed model and baselines including precision, recall and F1-score.

| Model | | Ctrl. (355) | | | Flu. (269) | | | No-Com. (31) | | | No-Flu. (49) | | | Weighted avg. (704) | | |
|---|---|---|---|---|---|---|---|---|---|---|---|---|---|---|---|---|
| | | Prec | Rec. | F1 | Prec | Rec. | F1 | Prec | Rec. | F1 | Prec | Rec. | F1 | Prec | Rec. | F1 |
| | SVM + Poly | 0.504 | 1.000 | 0.670 | 0.000 | 0.000 | 0.000 | 0.000 | 0.000 | 0.000 | 0.000 | 0.000 | 0.000 | 0.254 | 0.504 | 0.338 |
| Aphasia & | RF | 0.848 | 0.896 | 0.871 | 0.693 | 0.747 | 0.719 | 0.000 | 0.000 | 0.000 | 0.516 | 0.327 | 0.400 | 0.728 | 0.760 | 0.742 |
| Other Disorder | DT | 0.831 | 0.817 | 0.824 | 0.660 | 0.643 | 0.652 | 0.083 | 0.097 | 0.090 | 0.263 | 0.306 | 0.283 | 0.693 | 0.683 | 0.688 |
| Detection | LR | 0.779 | 0.766 | 0.773 | 0.611 | 0.807 | 0.696 | 0.000 | 0.000 | 0.000 | 0.000 | 0.000 | 0.000 | 0.627 | 0.695 | 0.655 |
| | AdaBoost | 0.928 | 0.839 | 0.882 | 0.629 | 0.699 | 0.662 | 0.000 | 0.000 | 0.000 | 0.257 | 0.367 | 0.303 | 0.726 | 0.716 | 0.719 |
| | MISA | 0.846 | 0.960 | 0.899 | 0.725 | 0.758 | 0.741 | 0.000 | 0.000 | 0.000 | 0.785 | 0.224 | 0.349 | 0.758 | 0.789 | 0.761 |
| Multimodal | MulT | 0.885 | 0.885 | 0.885 | 0.699 | 0.810 | 0.750 | 0.000 | 0.000 | 0.000 | 0.552 | 0.327 | 0.410 | 0.751 | 0.778 | 0.761 |
| | MAG | 0.914 | 0.775 | 0.838 | 0.625 | 0.792 | 0.698 | 0.000 | 0.000 | 0.000 | 0.482 | 0.551 | 0.514 | 0.733 | 0.732 | 0.725 |
| | SP-Trans. | 0.867 | 0.921 | 0.893 | 0.695 | 0.796 | 0.742 | 0.000 | 0.000 | 0.000 | 0.579 | 0.224 | 0.324 | 0.743 | 0.784 | 0.756 |
| **Ours** | | 0.949 | 0.949 | 0.949 | 0.799 | 0.885 | 0.840 | 0.176 | 0.097 | 0.125 | 0.647 | 0.449 | 0.530 | 0.837 | 0.852 | 0.842 |

- **Random Forest** (Cui et al., 2021; Balagopalan et al., 2020): The Random Forest classifier has demonstrated improved performance in various speech classification tasks, including speech/non-speech discrimination and speech emotion recognition.

- **Decision Tree** (Qin et al., 2018b; Balagopalan and Novikova, 2021): For the decision tree model, the text and audio embedding vectors of each subject are used as input. The model is trained using the gini criterion.

- **AdaBoost** (Cui et al., 2021): We set the maximum number of estimators to 50 and apply the boosting algorithm for AdaBoost.

**Multimodal fusion Baselines.** To accurately classify types of aphasia, we use multimodal data consisting of text, video, and audio. Our aim is to demonstrate the effectiveness of our model compared to existing multimodal sentiment analysis baseline models, highlighting its validity in the context of aphasia.

- **MulT** (Tsai et al., 2019): The MulT model is designed for analyzing multimodal human language. It utilizes a crossmodal attention mechanism to fuse multimodal information by directly attending to low-level features in other modalities.

- **MISA** (Hazarika et al., 2020): MISA operates two subspaces for each modality: a modality-invariant subspace that captures commonalities and reduces modality gaps, and a modality-specific subspace that captures distinctive features. These representations enable a holistic view of multimodal data, which is used for fusion and task predictions.

- **MAG** (Rahman et al., 2020b): MAG is a method introduced for efficient finetuning of large pre-trained Transformer models for multimodal language processing. It represents nonverbal behavior as a vector with trajectory and magnitude, allowing it to shift lexical representations within the pre-trained Transformer model.

- **SP-Transformer** (Cheng et al., 2021): SP-Transformer uses a sampling function to create a sparse attention matrix, reducing long sequences to shorter hidden states. The model captures interactions between hidden states of different modalities at each layer. To improve efficiency, it employs Layer-wise parameter sharing and Factorized Co-Attention, minimizing the impact on task performance.

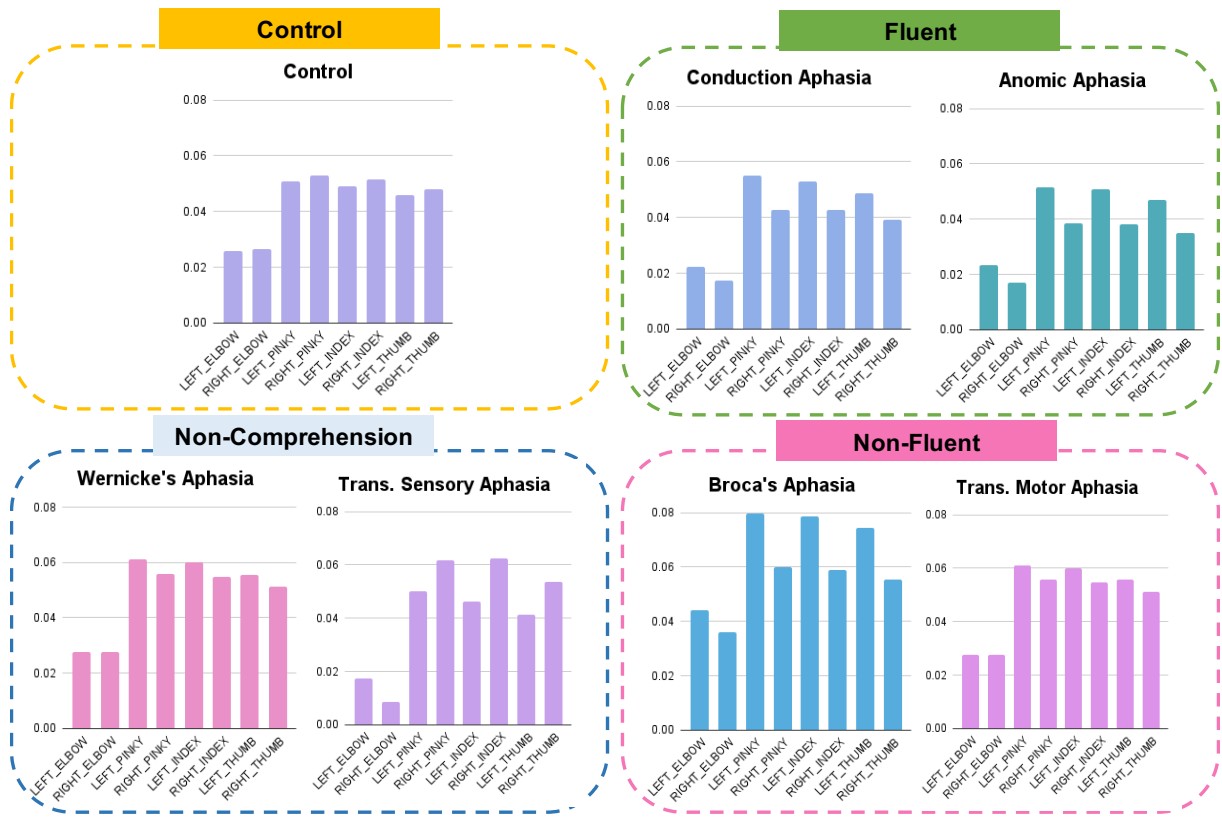

Figure 6: Visualization of the standard deviation values for pose key points of each aphasia type

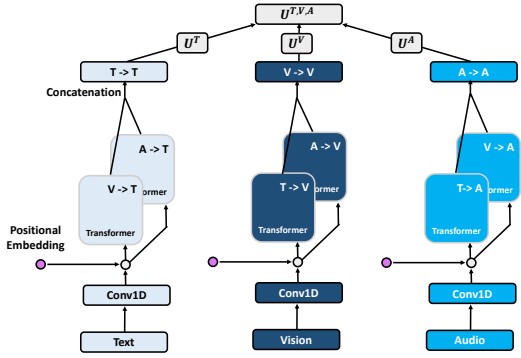

Figure 7: Architecture of Multimodal Transformer Encoder (Rahman et al., 2020b)