# OpenReview forum: "Learning Co-Speech Gesture for Multimodal Aphasia Type Detection"
_EMNLP/2023/Conference — EMNLP 2023 Main_

### Official Review · Reviewer_xF2x · 2023-08-02

**Typos Grammar Style And Presentation Improvements:** Not found
**Soundness:** 3

**Excitement:**

4: Strong: This paper deepens the understanding of some phenomenon or lowers the barriers to an existing research direction.

**Missing References:**

Not found

**Paper Topic And Main Contributions:**

In this paper, the authors use language and gesture information to detect aphasia types. They propose a new multimodal graph neural network for aphasia type detection. The authors also conduct an ablation study to assess the effectiveness of linguistic, acoustic, and gesture features. They find that gesture features outperform acoustic features. The authors also analyze the gender effect on aphasia type detection.

**Questions For The Authors:**

1. In this paper, author mentioned 3651 samples were included in the dataset, and each duration was approximately 23 seconds.
The number of samples in the dataset is a good starting point, but it is not enough to train a complex model like the one proposed in the paper. The authors should report the number of parameters in the final model to give readers a better understanding of its complexity.

2. The authors claim that gesture information outperforms acoustic information which looks fine to me, how about the text information compare to gesture information? Based on the table 4, it seems like text features outperform the gesture features, and the authors should provide more explanation for it.

**Reasons To Accept:**

1. This paper is novel in that it attempts to detect aphasia types rather than simply the existence of aphasia. This is an improvement over previous methods, as it allows for more accurate diagnosis and treatment of aphasia.

2. The authors compare the effectiveness of acoustic features and gesture features for aphasia detection. They find that gesture features are more effective than acoustic features, which is also a novel finding.

**Reasons To Reject:**

I am removing my previous comment because it does not obey the review policy. Instead, I am adding some questions to the following sections.

**Reproducibility:**

3: Could reproduce the results with some difficulty. The settings of parameters are underspecified or subjectively determined; the training/evaluation data are not widely available.

**Reviewer Confidence:**

3: Pretty sure, but there's a chance I missed something. Although I have a good feel for this area in general, I did not carefully check the paper's details, e.g., the math, experimental design, or novelty.

---

> ### Author Rebuttal · Authors · 2023-08-25
>
> We thank you for your valuable comments. We hope provided explanations can help to clarify misunderstandings.
>
> ## [Typo Errors]
> We will review the paper carefully to ensure no typos exist in the final version.
> - (Figure 1) The figure size and clarity should be increased.
> - (Table 4) :  the letters T,V and A in table 4 should be explained. (T: text / V: video (Gesture in this paper), / A: Audio)
> - (L554) ~ female participants. -> ~female participants (Table 5).
> - (L557) Table 1 -> Table 8 in Appendix
>
> ## [Contribution]
> Our interdisciplinary research brings valuable contributions to both the fields of machine learning and speech-language pathology. When it comes to diagnosing various types of aphasia, the traditional evaluation methods used for individuals with aphasia (PWA) are not only demanding in terms of time and expenses, but they also heavily rely on the expertise of pathologists.
>
> To tackle these difficulties, we suggest employing a graph neural network to create a model that automatically identifies different forms of aphasia. This model takes advantage of multiple sources of information, including co-speech gestures, audio, and text. Through this approach, our goal is to simplify the diagnostic process, alleviate the workload of pathologists, and simultaneously enhance the precision of detecting aphasia types.
>
> We summarize the contributions of this work as follows.
>
> 1. To the best of our knowledge, this is the first attempt that uses gesture and speech information for automatic aphasia types detection.
> 2. Through extensive experimentation, our model demonstrates superior performance compared to previous methods in identifying various aphasia types. The incorporation of Graph Neural Networks (GNN) proves particularly effective in generating multimodal features by capturing the interplay between speech and gesture data, resulting in improved performance.
> 3. Our analysis highlights the increased significance of gesture features over acoustic features in predicting aphasia types. This underscores the unique role of gesture expression as a distinguishing characteristic among individuals with different types of aphasia.
> 4. We illustrate that the qualitative analysis facilitated by our proposed model can offer clinicians a more comprehensive understanding of aphasia patients, aiding in the delivery of timely interventions.
> 5. We plan to make our code accessible to researchers within both the machine learning and pathology communities.
>
> ===========================================================
>
> # Aug 28th Additional Rebuttal
>
> Thank you for pointing out the settings of methods and our dataset.  We will organize and add the answers below to the final version for clarification.
>
> ## [Statistical significance of Performance Result]
> Results marked with an asterisk (*) indicate statistical significance compared to MAG (𝑝 < 0.05) according to the Wilcoxon’s signed rank test. Moreover, our observations indicate statistical significance (𝑝 < 0.05) in relation to MISA and MulT, the second-best performing models, assessed by the Wilcoxon’s signed rank test for both the overall average F1 score and Control (Type 0).
>
> |    Model   |                | Label (F1-score) |        |       |         |         |
> |:----------:|:--------------:|:----------------:|:------:|:-----:|:-------:|:-------:|
> |            |                |       Total      |  Ctrl  |  Flu  | Non-Com | Non-Flu |
> | Aphasia &  |   SVM + Poly   |       0.338      |  0.670 | 0.000 |  0.000  |  0.000  |
> |  Dementia  |       RF       |       0.742      |  0.871 | 0.719 |  0.000  |  0.400  |
> |  Detection |       DT       |       0.688      |  0.824 | 0.652 |  0.090  |  0.283  |
> |            |       LR       |       0.655      |  0.773 | 0.696 |  0.000  |  0.000  |
> |            |    AdaBoost    |       0.719      |  0.882 | 0.662 |  0.000  |  0.303  |
> |   Fusion   |      MISA      |       0.761      |  0.899 | 0.741 |  0.000  |  0.349  |
> |            |      MulT      |       0.761      |  0.885 | 0.750 |  0.000  |  0.410  |
> |            |       MAG      |       0.725      |  0.838 | 0.698 |  0.000  |  0.514  |
> |            | SP-Transformer |       0.756      |  0.893 | 0.742 |  0.000  |  0.324  |
> |    Ours    |    Chunk 30    |       0.732      |  0.906 | 0.676 |  0.303  |  0.446  |
> |            |    Chunk 50    |      0.842*      | 0.949* | 0.84* |  0.125  |  0.530  |
>
> ## [Data size & Model Complexity]
> We acknowledge your concerns about the dataset's size being potentially insufficient for training a complex model. To ensure robust performance, we conducted our experiments using a Group Stratified 5-fold cross-validation approach. In the table below, we outline the complexity of our model. It's worth noting that when compared to other studies focused on detecting cognitive disorders [1-3], our dataset is notably larger—more than 2-3 times in size. Notably, the ADReSS dataset, a significant benchmark in Dementia Detection [1], comprises 156 users across the entire dataset, while our study includes 507 users sourced from various corpora (as detailed in Table 10 in the Appendix). Moreover, the performance results we presented in the table above demonstrate that our model statistically outperforms all baseline models.
>
> |             **Type**             | **Params** |
> |:--------------------------------:|:----------:|
> |              Dropout             |      0     |
> |           RobertaModel           |    124 M   |
> |          HeteroGraphConv         |   24.9 M   |
> |               LSTM               |    192 K   |
> |               LSTM               |   25.8 K   |
> |             Attention            |     138    |
> |             Attention            |     50     |
> |         MulT Transformer         |    3.3 M   |
> |              Linear              |    295 k   |
> |              Linear              |    1.5 k   |
> |         Trainable params         |    153 M   |
> |       Non-trainable params       |      0     |
> |           Total params           |    153 M   |
> | Total estimated params size (MB) |   613.536  |
>
> - [1]https://dementia.talkbank.org/ADReSS-2021/
> - [2]  Li, C., Knopman, D., Xu, W., Cohen, T., & Pakhomov, S. (2022, May). GPT-D: Inducing Dementia-related Linguistic Anomalies by Deliberate Degradation of Artificial Neural Language Models. In Proceedings of the 60th Annual Meeting of the Association for Computational Linguistics (Volume 1: Long Papers) (pp. 1866-1877).
> - [3] Farzana, S., Deshpande, A., & Parde, N. (2022, May). How you say it matters: Measuring the impact of verbal disfluency tags on automated dementia detection. In Proceedings of the 21st Workshop on Biomedical Language Processing (pp. 37-48).
>
>
> ## [Analysis on different modalities]
> We have already provided an explanation indicating that linguistic features offer a higher level of information for the identification of language impairment disorders, which is consistent with findings from previous clinical research (L497).
>
> Moreover, our analysis in Table 4 focuses on different modalities to determine which—acoustic, gesture, or textual features—better complement the others in detecting aphasia types. While many studies in the domain of linguistic pathology typically concentrate solely on speech data, which includes linguistic and acoustic elements, we have observed that gesture information holds significant potential in identifying aphasia types. However, it is noteworthy that incorporating all three modalities yields the most optimal performance. This finding highlights that combining linguistic features with visual and acoustic attributes, as well as understanding their interrelationships, proves more effective than relying solely on one modality.

---

### Official Review · Reviewer_kFrs · 2023-08-05

**Soundness:** 4

**Excitement:**

4: Strong: This paper deepens the understanding of some phenomenon or lowers the barriers to an existing research direction.

**Paper Topic And Main Contributions:**

In this paper, the novel graph neural network for aphasia type detection is presented. This work addresses the problem of a detailed diagnosis of aphasia type, which is important in clinical settings for proper treatment procedures. The proposed model makes use of multiple modalities, including acoustics of human speech, text in the format of transcribed speech, and corresponding gesture patterns. Multimodal features are generated with the help of a graph neural network for each aphasia type, enabling the acquisition of rich semantics across modalities. The model learns the correlation between the speech and gesture modalities for multiple aphasia types, and it helps it achieve SOTA results and outperform models relying on any single modality.
The method of applying a graph neural network to capture interconnections across multiple modalities and the presented novel model for aphasia type detection are two main contributions of this work.



**Questions For The Authors:**

1. It would be beneficial to see if the best performance results (Tables 2,3,4 and 5) are statistically significantly different from the rest of models you compare to.
2. Explain the letters T,V and A in table 4
3. Looks like Table 5 was never refered to in the paper

**Reasons To Accept:**

This is a well-written paper that presents a sound work with some interesting and useful contributions to the NLP community. The proposed model is explained in sufficient details and the corresponding code is promised to be shared with the community. Experimental work presented in the paper is technically sound and detailed - model performance is compared to multiple strong baselines, several ablation studies give a good understanding of the impact of individual model components, impact of different modalities, impact of gender, and the effectiveness of the proposed Speech-Gesture Graph Encoder. In addition to this quantitative analysis, the qualitative analysis is performed on four representative cases involving different types of aphasia. All the claims made in this work are strongly supported by empirical evidences.

**Reasons To Reject:**

I see no strong reasons to reject this paper. There are several minor weaknesses that I outline in the section "Questions for the Authors" but I believe that the paper is strong enough in its current version already and provides substantial contributions to be accepted to the conference.

**Reproducibility:**

3: Could reproduce the results with some difficulty. The settings of parameters are underspecified or subjectively determined; the training/evaluation data are not widely available.

**Reviewer Confidence:**

4: Quite sure. I tried to check the important points carefully. It's unlikely, though conceivable, that I missed something that should affect my ratings.

---

> ### Author Rebuttal · Authors · 2023-08-25
>
> We thank you for your invaluable suggestions and positive comments! We plan to organize and revise the answers below so there are no unclear explanations in the final version.
>
> ## [Typo Errors]
> We will review the paper carefully to ensure no typos exist in the final version.
> - (Figure 1) The figure size and clarity should be increased.
> - (Table 4) :  the letters T,V and A in table 4 should be explained. (T: text / V: video (Gesture in this paper), / A: Audio)
> - (L554) ~ female participants. -> ~female participants (Table 5).
> - (L557) Table 1 -> Table 8 in Appendix
>
> ## [Statistical significance of performance]
> Results marked with an asterisk (*) indicate statistical significance compared to MAG (𝑝 < 0.05) according to the Wilcoxon’s signed rank test. Moreover, our observations indicate statistical significance (𝑝 < 0.05) in relation to MISA and MulT, the second-best performing models, assessed by the Wilcoxon’s signed rank test for both the overall average F1 score and Control (Type 0).
>
>
> |    Model   |                | Label (F1-score) |        |       |         |         |
> |:----------:|:--------------:|:----------------:|:------:|:-----:|:-------:|:-------:|
> |            |                |       Total      |  Ctrl  |  Flu  | Non-Com | Non-Flu |
> | Aphasia &  |   SVM + Poly   |       0.338      |  0.670 | 0.000 |  0.000  |  0.000  |
> |  Dementia  |       RF       |       0.742      |  0.871 | 0.719 |  0.000  |  0.400  |
> |  Detection |       DT       |       0.688      |  0.824 | 0.652 |  0.090  |  0.283  |
> |            |       LR       |       0.655      |  0.773 | 0.696 |  0.000  |  0.000  |
> |            |    AdaBoost    |       0.719      |  0.882 | 0.662 |  0.000  |  0.303  |
> |   Fusion   |      MISA      |       0.761      |  0.899 | 0.741 |  0.000  |  0.349  |
> |            |      MulT      |       0.761      |  0.885 | 0.750 |  0.000  |  0.410  |
> |            |       MAG      |       0.725      |  0.838 | 0.698 |  0.000  |  0.514  |
> |            | SP-Transformer |       0.756      |  0.893 | 0.742 |  0.000  |  0.324  |
> |    Ours    |    Chunk 30    |       0.732      |  0.906 | 0.676 |  0.303  |  0.446  |
> |            |    Chunk 50    |      0.842*      | 0.949* | 0.84* |  0.125  |  0.530  |

---

### Official Review · Reviewer_q4Gj · 2023-08-12

**Soundness:** 3

**Excitement:**

4: Strong: This paper deepens the understanding of some phenomenon or lowers the barriers to an existing research direction.

**Paper Topic And Main Contributions:**

In this paper the authors propose a graph neural network based framework that uses information from speech and gesture modalities for Aphasia type detection. The main contributions of the paper are the multimodal GNN based framework for Aphasia type detection, it's comparison with existing benchmark models on a standard dataset and the evaluation of the impact of the contributing modalities. The authors claim that this framework is the first to leverage both speech and gesture information.

**Questions For The Authors:**

The authors should report ASR WER performance in the paper.

The authors should correct the line ... "Unlike previous ASR systems focusing on transcribing clean speech, Whisper can capture filler words ..." -> Most traditional ASR systems are able to well recognize disfluencies as long as they are trained using data that contains disfluencies. Also the claim that Whisper can capture all disfluencies should be verified.

**Reasons To Accept:**

Given that there is limited work on investigation of models for automatically detecting Aphasia types, this topic is relatively novel and will be of great interest to the community.
The proposed GNN model using multimodal information from speech and gesture data is novel. The proposed model outperforms existing state-of-the-art models by quite a margin setting a new standard for state-of-the-art models.
The experiments and the quality analysis of the results are thorough.

**Reasons To Reject:**

The authors mention they used Whisper for ASR transcription. To the best of my knowledge, Whisper is well known to filter out disfluencies and filler words in the transcription or generate very limited disfluencies. In the absence of the ASR word error rate performance report, it is hard to justify the impact of disfluency keywords on Aphasia type detection.

**Reproducibility:**

3: Could reproduce the results with some difficulty. The settings of parameters are underspecified or subjectively determined; the training/evaluation data are not widely available.

**Reviewer Confidence:**

4: Quite sure. I tried to check the important points carefully. It's unlikely, though conceivable, that I missed something that should affect my ratings.

**Typos Grammar Style And Presentation Improvements:**

The gesture depiction in Figure 1 is very hard to read for a reader and the authors should consider increasing the figure size and clarity.

---

> ### Author Rebuttal · Authors · 2023-08-25
>
> We thank you for your comments on the experiments. Based on your comments, we further conducted additional experiments. The results of these experiments are included in the answer below and will be added to the final version.
>
> ## [Typo Errors]
> We will review the paper carefully to ensure no typos exist in the final version.
> - (Figure 1) The figure size and clarity should be increased.
> - (Table 4) :  the letters T,V and A in table 4 should be explained. (T: text / V: video (Gesture in this paper), / A: Audio)
> - (L554) ~ female participants. -> ~female participants (Table 5).
> - (L557) Table 1 -> Table 8 in Appendix
>
> ## [Whisper Disfluency]
> As you mentioned earlier, Whisper is highly effective in identifying and removing disfluencies and filler words from transcriptions. In other words, it excels at detecting these linguistic elements, and you can effortlessly extract such words using the straightforward options provided by the Whisper API.
>
> For a more accurate evaluation, we have reported the Word Error Rate (WER) results along with ASR examples below.
>
> |  Type  |                        | Average WER |                    |
> |:------:|:----------------------:|:-----------:|:------------------:|
> |        |                        |   Raw text  | Preprocessing text |
> | type 0 |         Control        |    0.332    |        0.180       |
> | type 1 |   Anomic/ Conduction   |    0.714    |        0.623       |
> | type 2 | Wernicke/ TransSensory |    0.607    |        0.542       |
> | type 3 |    Broca/ TransMotor   |    0.967    |        0.974       |
> |  total |                        |    0.615    |        0.521       |
>
>
> The results showed that the average WER for all participants was similar to previous research, with a decrease in WER for less fluent aphasia types (e.g., type 3) [1].
>
> Furthermore, we recognized the considerable impact of preprocessing on WER (i.e., raw text vs. preprocessing text), stemming from the prevalence of fillers and disfluencies within the texts of individuals with language disorders.
>
> Nevertheless, our real-world examples show that the ASR system can capture not only the overall context, but also fillers and disfluencies.
> Hence, considering the implications for individuals with language disorders, exploring innovative methods for conducting disfluency ASR assessments in future research might be valuable.
>
> [1] Weiner, J., Engelbart, M., & Schultz, T. (2017, August). Dementia Detection from Speech in Manual and Automatic Transcriptions. In Interspeech (pp. 3117-3121).
>
>
> |    | Raw                                                                                                                                                      | Raw_preprocessing                                                                                | Whisper with Disfluency                                                                                                                                                  |
> |----|----------------------------------------------------------------------------------------------------------------------------------------------------------|--------------------------------------------------------------------------------------------------|--------------------------------------------------------------------------------------------------------------------------------------------------------------------------|
> |  1 | &-um the [//] &-um a [/] a very wicked &-um woman and her daughters &=ges:two two of them were &-um +...                                                 | a very wicked woman and her daughters two of them were...                                        | [\*] the, [\*] a, [\*] a [\*] very wicked [\*] woman [\*] and [\*] her [\*] daughters, two of [\*] them [\*] were                                                                  |
> |  2 | and Cinderella she was <a &-um daughter &-um of> [//] &-um a stepdaughter . and &-um they didn\'t like her .                                             | and Cinderella she was a stepdaughter. and they didn't like her.                                 | [\*] Anne[\*] Cinderella. [\*] She [\*] was [\*] a [\*] daughter [\*] of [\*] a [\*] step-daughter. And [\*] they [\*] didn't [\*] like her,                                         |
> |  3 | the two &-um &-um daughters and the stepmother .                                                                                                         | the two daughters and the stepmother .                                                           | the [\*] two [\*] daughters and[\*] the [\*] step-mothers.                                                                                                                   |
> |  4 | so ‡ they treat her very badly .                                                                                                                         | so they treat her very badly.                                                                    | So [\*] they [\*] treat her [\*] very [\*] badly.                                                                                                                            |
> |  5 | and she talked to the birds and trees and +...                                                                                                           | and she talked to the birds and trees and...                                                     | And [\*] she [\*] talked to [\*] the [\*] birds and trees [\*] and                                                                                                            |
> |  6 | she said +"/.  +" oh ‡ <I wonder> [//] I wish I was <going to the king &-um no> [//] &-um going to the palace and all of that but I\'ll wait .           | she said oh "I wish I was going to the palace and all of that but I'll wait."                    | she, [\*]I wonder, I wish I was [\*] going to [\*] the [\*] king, [\*] going to the palace [\*] and [\*] all of that, [\*] but [\*] I would [\*] wait,                             |
> |  7 | okay . [+ exc]  and &-um a long &+ag &-um time ago &-um <a thing> [//] &-um a note came from the palace and it said &-um they are going to have a ball . | okay. and a long time ago a note came from the palace and it said they are going to have a ball. | okay, [\*] and [\*] a long [\*] time [\*] ago, [\*] a [\*] thing, [\*] a [\*] note came [\*] from the [\*] palace and [\*] it [\*] said[\*] they are going to [\*] have [\*] a [\*] ball |
> |  8 | and sɪndəwɛlə@u [: Cinderella] [* p:n] said +"/.  +" can I go ?                                                                                          | and Cinderella said "can I go?"                                                                  | and [\*] Cinderella said, can I [\*] go                                                                                                                                    |
> |  9 | and the mother said +"/.  +" no you can\'t go .  +" &=finger:wag and you are going to stay here and keen [: clean] [* p:w] the house .                   | and the mother said. "no you can't go. and you are going to stay here and clean the house."      | and the [\*] mother said, no, you [\*] can't go and you[\*] are going to [\*] stay here and [\*] clean the [\*] house.                                                         |
> | 10 | &=sighs sɪndəwɛlə@u [: Cinderella] [* p:n] was sad .  +" I can\'t go to the ball &=laughs .                                                              | Cinderella was sad "I can't go to the ball."                                                     | Cinderella [\*] was sad, [\*] I [\*] can't go to the [\*] ball.                                                                                                              |

---

### Meta-Review · Area_Chair_aj9X · 2023-09-04

**Recommendation:** 4
**Confidence:** 4

**Metareview:**

Quality:

The paper presents high-quality results, which is noted by the reviewers as follows:

-	“The proposed model outperforms existing state-of-the-art models by quite a margin setting a new standard for state-of-the-art models.”

-	“The experiments and the quality analysis of the results are thorough.”

-	“Experimental work presented in the paper is technically sound and detailed - model performance is compared to multiple strong baselines, several ablation studies give a good understanding of the impact of individual model components, impact of different modalities, impact of gender, and the effectiveness of the proposed Speech-Gesture Graph Encoder. In addition to this quantitative analysis, the qualitative analysis is performed on four representative cases involving different types of aphasia. All the claims made in this work are strongly supported by empirical evidences.”

Clarity:

The paper is clear and easy to understand: “This is a well-written paper that presents a sound work with some interesting and useful contributions to the NLP community. The proposed model is explained in sufficient details and the corresponding code is promised to be shared with the community.”

Originality:

The work is highly original in that it uses a GNN model with multimodal information from speech and gesture data to detect different types of aphasia and provides evidence on the importance of acoustic features.

Significance:

Finally, the work is significant in that it presents a multimodal study on aphasia detection, making a contribution towards a holistic approach towards detecting the condition.

The rebuttal provided by the authors in response to the questions of the reviewers is substantial.

---

### Decision · Program_Chairs · 2023-10-07

**Decision:**

Accept-Main

**Comment:**

Quality:

The paper presents high-quality results, which is noted by the reviewers as follows:

-	“The proposed model outperforms existing state-of-the-art models by quite a margin setting a new standard for state-of-the-art models.”

-	“The experiments and the quality analysis of the results are thorough.”

-	“Experimental work presented in the paper is technically sound and detailed - model performance is compared to multiple strong baselines, several ablation studies give a good understanding of the impact of individual model components, impact of different modalities, impact of gender, and the effectiveness of the proposed Speech-Gesture Graph Encoder. In addition to this quantitative analysis, the qualitative analysis is performed on four representative cases involving different types of aphasia. All the claims made in this work are strongly supported by empirical evidences.”

Clarity:

The paper is clear and easy to understand: “This is a well-written paper that presents a sound work with some interesting and useful contributions to the NLP community. The proposed model is explained in sufficient details and the corresponding code is promised to be shared with the community.”

Originality:

The work is highly original in that it uses a GNN model with multimodal information from speech and gesture data to detect different types of aphasia and provides evidence on the importance of acoustic features.

Significance:

Finally, the work is significant in that it presents a multimodal study on aphasia detection, making a contribution towards a holistic approach towards detecting the condition.

The rebuttal provided by the authors in response to the questions of the reviewers is substantial.